# Phase Separation of SARS-CoV-2 Nucleocapsid Protein with TDP-43 Is Dependent on C-Terminus Domains

**DOI:** 10.3390/ijms25168779

**Published:** 2024-08-12

**Authors:** Michael J. Strong, Crystal McLellan, Brianna Kaplanis, Cristian A. Droppelmann, Murray Junop

**Affiliations:** 1Molecular Medicine Group, Robarts Research Institute, Schulich School of Medicine & Dentistry, Western University, London, ON N6A 3K7, Canada; cwhitfi@uwo.ca (C.M.); cristiandroppelmann@gmail.com (C.A.D.); 2Department of Clinical Neurological Sciences, Schulich School of Medicine & Dentistry, Western University, London, ON N6A 3K7, Canada; 3Department of Pathology and Laboratory Medicine, Schulich School of Medicine & Dentistry, Western University, London, ON N6A 3K7, Canada; 4Department of Biochemistry, Schulich School of Medicine & Dentistry, Western University, London, ON N6A 3K7, Canada; bkaplani@uwo.ca (B.K.); mjunop@uwo.ca (M.J.)

**Keywords:** neurodegeneration, biomolecular condensates, nucleocapsid protein, RNA binding proteins, amyotrophic lateral sclerosis

## Abstract

The SARS-CoV-2 nucleocapsid protein (N protein) is critical in viral replication by undergoing liquid–liquid phase separation to seed the formation of a ribonucleoprotein (RNP) complex to drive viral genomic RNA (gRNA) translation and in suppressing both stress granules and processing bodies, which is postulated to increase uncoated gRNA availability. The N protein can also form biomolecular condensates with a broad range of host endogenous proteins including RNA binding proteins (RBPs). Amongst these RBPs are proteins that are associated with pathological, neuronal, and glial cytoplasmic inclusions across several adult-onset neurodegenerative disorders, including TAR DNA binding protein 43 kDa (TDP-43) which forms pathological inclusions in over 95% of amyotrophic lateral sclerosis cases. In this study, we demonstrate that the N protein can form biomolecular condensates with TDP-43 and that this is dependent on the N protein C-terminus domain (N-CTD) and the intrinsically disordered C-terminus domain of TDP-43. This process is markedly accelerated in the presence of RNA. In silico modeling suggests that the biomolecular condensate that forms in the presence of RNA is composed of an N protein quadriplex in which the intrinsically disordered TDP-43 C terminus domain is incorporated.

## 1. Introduction

Although the long-term repercussions of the COVID-19 pandemic remain to be fully defined, it is clear that a significant proportion of individuals exposed to the SARS-CoV-2 virus and who develop its clinical correlate COVID-19 will also develop one or more neurological sequelae as a manifestation of the post-COVID-19 condition (PCC) [1,2,3]. Amongst these consequences, there is growing interest in the potential role of the virus in driving the process of neurodegeneration by participating in pathological biomolecular condensate formation [1,4]. The latter, often manifesting as neuronal or glial cytoplasmic inclusions (NCIs and GCIs, respectively), is a neuropathological correlate of a broad range of adult-onset neurodegenerative disorders that includes amyotrophic lateral sclerosis (ALS), Alzheimer’s disease, and Parkinson’s disease [5,6,7,8].

SARS-CoV-2 forms a virion containing its genomic RNA (gRNA) with four structural proteins including the spike proteins (S1 and S2 subunits), the membrane protein (M) which facilitates assembly in the endoplasmic reticulum, the ion channel envelope protein (E protein), and the nucleocapsid protein (N protein). The latter is an RNA binding protein (RBP) that is highly conserved across coronaviruses and assembles with gRNA to form a helical ribonucleoprotein (RNP) complex through its propensity to form biomolecular condensates through liquid–liquid phase separation (LLPS) [9,10,11,12,13,14,15,16]. Encoded by the ninth ORF of the SARS-CoV-2 virus, the N protein is a 46 kDa protein containing 419 amino acid residues that is organized into an N-terminus domain (N-NTD) which contains the primary RNA binding domain (RBD), a C-terminus domain (N-CTD) containing a dimerization domain that is also capable of weaker RNA interactions while being the primary driver of phase separation, and a central Ser/Arg (SR)-rich flexible linker region (LKR) with several phosphorylation sites and which also has weak RNA binding capacity (Figure 1) [17,18,19,20,21,22]. Approximately 40% of the N protein is intrinsically disordered, including intrinsically disordered regions (IDRs) located at the N-terminus (amino acids 1–43), the C-terminus (amino acids 365–419), and within the LKR (Figure 1) [23]. The N protein self-assembles into higher-order oligomeric structures such that two N-terminus domains project from a tight core domain formed by dimerization of the N-CTDs [18]. The N protein dimer undergoes domain compaction by a process in which shorter RNA fragments (<30 nt) interact only with the N-CTD, while longer RNA fragments (>50–60 nt) are able to also interact with both N-NTDs, and in doing so, drives compaction of the N protein dimer into higher-order oligomeric structures which, in the presence of RNA, enhances biomolecular condensate formation [18,24,25,26].

Liquid–liquid phase separation (LLPS) of the N protein is dependent on the N-CTD and can be accelerated by the presence of full-length gRNA as well as homopolymeric RNAs (polyA, polyU, polyC, and polyG) [9,13,22,29,30]. Similar to the dynamics described in the context of RBPs in general, higher RNA concentrations suppress LLPS, consistent with a model of “reentrant RNA-mediated phase separation”. The N protein is extensively phosphorylated in its C-terminus domain while exhibiting minimal RBD post-translational modifications, which is unusual for RBPs [28]. However, the LKR is highly phosphorylated which affects RNA-induced LLPS and nucleocytoplasmic shuttling [12,21].

While the N protein plays a fundamental role in viral RNA replication and transcription through its incorporation in an RNP, it can also induce profound alterations in the immune response by suppressing the PKR-eIF2α-G3BP1/2 pathway of stress granule (SG) formation through direct interactions with the protein kinase PKR and G3BP1 [27,31,32,33]. Both G3BP1 and TIA-1 are key initiators of the condensation of RNA and proteins into stress granules. By directly interacting with the N-terminus nuclear transport factor 2 (NTF2)-like domain of G3BP1 (residues 11–134), aa F17 of the N protein fundamentally disrupts this assembly—potentially accounting for the significant reduction in SGs in SARS-CoV-2-infected cells [29,31,32,34,35,36,37]. When methylated at R95, further suppression of the SG response is observed [27]. While these observations speak to inhibition of the formation of the SG, it has also been suggested that the N protein can induce the disassembly of existent granules [7]. The N protein appears to be unique amongst coronaviruses in also being able to mediate processing body (PB) dissolution, further contributing to impairments in the host’s stress response in the presence of viral infection [38]. It remains to be determined whether these combined processes lead to the failure of RNA to be sequestrated into either SGs or PBs and thus to enhanced levels of “uncoated” mRNA which can then interact with the N protein to accelerate its phase separation.

Fundamental disturbances in RNA metabolism are a major contributor to the pathophysiology of ALS, as evidenced by the presence of pathological glial and neuronal cytoplasmic inclusions (NCIs) of RBPs, alongside fundamental alterations in RNA biogenesis [39,40]. A defining feature of RBPs is that they are highly intrinsically disordered, significantly exceeding the degree of intrinsic disorder of other human proteins, although their RNA binding domains (RBDs) are generally more structured. It is not surprising therefore that NCIs within individual degenerating motor neurons contain multiple RBPs forming heterogeneous fibrils and aggregates [41].

We have postulated that amongst the long-term consequences of SARS-CoV-2 infection, ongoing neuronal expression of the N protein may contribute to pathological biomolecular condensate formation by interacting with other endogenous intrinsically disordered proteins, RNA, or RBPs and may thus alter host cell RNA homeostasis [1]. In this study, we have examined how the presence of RNA impacts the ability of TAR DNA binding protein 43 (TDP-43), the most prevalent ALS-associated RBP observed to be a component of pathological NCIs, to undergo LLPS with the N protein. We have further examined the in silico, in vitro, and in vivo evidence that suggests that the expression of the N protein in the central nervous system, and specifically within neurons, could be considered a pathway of induction of a neurodegenerative disease state. While an attractive postulate, there however remain many unanswered questions as to the mechanism(s) by which the SARS-CoV-2 virus could directly give rise to such disease states.

## 2. Results

### 2.1. Protein Aggregation Assay

The ability of the SARS-CoV-2 N protein to form protein aggregates in vitro in the presence of RNA and TDP-43 was assessed using an aggregation detection reagent that fluoresces upon binding to non-specific protein aggregates. Aggregation reactions were performed with purified proteins and RNA oligonucleotides in turbidity/crowding buffer immediately before addition to the aggregation detection reagent. Consistent with the observations of others, the N protein alone was able to self-aggregate at concentrations from 1–2.5 µM up to 20–25 µM. The N protein was then added to either polyrA, polyrAUG, or polyrG RNA oligos to confirm the reported evidence that RNA stimulates N protein aggregation (Figure 2). These specific RNA oligonucleotides were selected on the basis of published studies demonstrating their ability to induce SARS-CoV-2 N protein phase separation [9,42]. The assay determined that when the N protein was added to each RNA oligo, protein aggregation was induced above the aggregation level of RNA alone. PolyrG RNA induced further aggregation of the N protein beyond that of the N protein alone, consistent with previous reports [9,42].

The N protein was also added to either GST-tagged full-length or N-terminus TDP-43 (FL-TDP-43 and TDP-43^1–269^, respectively) at a molar ratio of 1:0.8. The presence of these proteins did not significantly change the aggregation status of 1 or 2.5 µM of the N protein (Figure 2A). However, when the N protein, polyrA RNA, and FL-TDP-43 were added prior to aggregation detection, there was an increase in aggregation beyond the level determined for the N protein and polyrA alone. An increase in aggregation was also detected when the N protein, polyrG, and FL-TDP-43 were mixed, beyond that of the N protein alone. However, the bulk of this aggregation was also detected in the presence of the N protein and polyrG. When TDP-43^1–269^ was added to the N protein and polyrG at an equal concentration used for FL-TDP-43, the aggregation level was attenuated and did not increase beyond N protein alone levels. When TDP-43^1–269^ was used at a 5-fold higher concentration compared to FL-TDP-43, the aggregation results were similar to those of FL-TDP-43, but still attenuated, particularly at later time points in the time course (>30 min) (Appendix A). polyrAUG RNA did not show different results compared to polyrA when used in the aggregation reactions and thus was not tested in repeated experiments (Appendix A). The time course evaluations (50 min) did not show the re-entry phenomenon for any of the combinations (Appendix A). Efforts to undertake similar studies using the TDP-43 C-terminus domain were unsuccessful due to our inability to maintain this protein in a solubilized state.

Together, these results confirm that the N protein can self-aggregate and that this aggregation is enhanced in the presence of polyrG RNA. In addition, the N protein mixed with polyrA RNA and FL-TDP-43 can induce aggregation compared to the N protein with polyrA RNA, and the presence of polyrG RNA rather than polyrA RNA further increases this aggregation. However, the N protein with polyrG RNA and TDP-43^1–269^ does not induce an increase in aggregation, supporting the notion that the C-terminus domain of TDP-43 is key to N protein TDP-43 molecular condensate induction.

### 2.2. Surface Plasmon Resonance

Surface plasmon resonance (SPR) was used to quantify the in vitro interaction between the SARS-CoV-2 N protein and TDP-43. The N protein was fixed to a carboxymethyl dextran chip surface as the immobilized ligand and TDP-43 was injected as the potential binding partner. GST-tagged full-length TDP-43 or N-terminus TDP-43 (without the GST tag) were injected onto the chip at the indicated concentrations. The SPR experiments showed a direct interaction between the N protein and both the full-length TDP-43 and N-terminus TDP-43 with a K_D_ of 4.88 ± 1.13 μM and >167 μM, respectively, using steady-state analysis (Figure 3A,B). The interaction between the N protein and full-length TDP-43 was not dependent on the GST tag as GST alone at concentrations of 2–20 μM did not show any interaction with the N protein and the untagged full-length TDP-43 protein showed robust interaction with the N protein (K_D_ 125 ± 41.6 nM) (Appendix A). These studies suggest that the TDP-43 C-terminus domain is a major determinant of the interaction between the N protein and TDP-43. Unfortunately, this latter domain is soluble in urea but not otherwise, precluding its use in the SPR study.

### 2.3. In Silico Modeling of N Protein, TDP-43, and RNA Heteropolymers

To further understand the interaction between the N protein and FL-TDP-43, we used in silico modeling (AlphaFold 3) to evaluate their ability to form biomolecular condensates [43]. When modeling the N protein as a quadriplex in the presence of two FL-TDP-43 proteins, AlphaFold3 predicts a symmetrical complex with TDP-43 positioned more peripherally (Figure 4A). Here, RNA binding sites on both proteins appeared to be involved in the complex interaction, while the intrinsically disordered C-termini of TDP-43 remained predominately unincorporated (Figure 4A) (Appendix A). In contrast, upon the addition of either a random 20-mer or 70-mer RNA oligo, condensation occurred with the intrinsically disordered TDP-43 C-terminal domains integrated within the core of the heteropolymer complex, adopting helices in structure (Figure 4B). This was most marked when modeled with the 70-mer oligos in that there appeared to be alignment between TDP-43 C-terminus helices and the RNA binding region within the N-CTD of the N protein, and the 70-mer oligo was interwoven throughout (Figure 4C) (Appendix A). Meanwhile, the N-terminus RRM1/RRM2-containing domain of TDP-43 appeared to be externally facing and ‘cap’ the top of the N protein quadriplex. This complex appeared to be more densely compacted compared to the protein complex predictions in the absence of RNA. In addition to this, the total surface area of the IDR of TDP-43 chains were 4.4 times more buried than in the absence of RNA (Appendix A).

In summary, in silico modeling predicts that TDP-43 and the N protein will form a complex which, in the presence of RNA, becomes a more densely packed heteropolymer in which interactions are mediated through the N-CTD and the IDR of the C-terminus of TDP-43.

## 3. Discussion

We have demonstrated using both in silico projections and co-assembly studies (SPR and aggregation assays) that the N protein and TDP-43 can, ex vivo, form a biomolecular complex. Our SPR data suggest that the C-terminus domain of TDP-43 is the major determinant of the interaction between TDP-43 and the N protein with a K_D_ for the full-length TDP-43 interaction of 4.88 ± 1.13 μM, whereas that for the isolated N-terminus domain containing both RRMs is greater than 167 μM. In addition, our aggregation data suggest that the presence of RNA will further enhance this process. Given the use of the full-length recombinant N protein in our assays, we cannot ascertain whether the interaction is driven by the N-CTD, the N-NTD, or the full-length N protein. However, given previous observations regarding the role of RNA in mediating N protein heteropolymer formation [18], the enhancement of LLPS observed in the aggregation assay in the presence of short RNA oligomers would be consistent with the interaction being dependent on the N-CTD. While preliminary, in silico predictions suggest that this is due to the formation of a complex biomolecular condensate consisting of an N protein quadriplex interacting with the intrinsically disordered domains of two TDP-43 molecules and that this is dependent on the presence of RNA. There are however well-described limitations to the use of in silico modeling to characterize complex multiple-molecule interactions including that such predictions are performed in the absence of an ability to account for post-translational modifications (vide infra) [44,45,46]. Thus, while we have proposed an attractive model, subsequent validation will be required.

### 3.1. The Role of RNA Binding Proteins in Biomolecular Condensate Formation

It is now understood that intracellular LLPS giving rise to membraneless organelles (MLOs) is a critical physiological process that results in the formation of dynamic, highly functional biomolecular condensates. In contrast to more classical canonical macromolecular particles (e.g., RNA), biomolecular condensates are highly dynamic and readily exchange components with their surroundings, in part due to the relatively weak, non-covalent nature of interactions within the condensates [47]. They can range from 20 nm (i.e., interchromatin granules) to 1–6 μm (i.e., P bodies) in diameter [48]. Because of the fluidity of their composition, biomolecular condensates can display regions of differing composition and density, potentially underlying their ability to drive independent processes within the same condensate [49].

The most prevalent biomolecular condensates are combined RNA and protein assemblies (ribonucleoprotein (RNP) granules, or RNP bodies) such as SGs and PBs, where assembly/disassembly is driven by the phase separation process. Also included in this grouping are nucleoli, nuclear paraspeckles, Cajal bodies, transport granules, and Balbiani bodies [50]. The key physiological functions of biomolecular condensates include major aspects of RNA metabolism, including transcription, pre-mRNA processing, subcellular localization, and the regulation of translation and decay [48,51].

The process of biomolecular condensation is divided into two broad categories of participating proteins: scaffold proteins that drive reversible condensate formation or clients which are proteins that preferentially partition into condensates [52]. The predominant proteins found within biomolecular condensates are RNA binding proteins (RBPs), a class of *trans*-acting regulatory proteins that interact with *cis*-acting RNA elements, with their greatest specificity being for mRNA elements [53,54]. RBPs interact with cognate transcripts through a small repertoire of RNA binding domains (RBDs) that include RNA recognition motifs (RRMs; typically, an average of 90 amino acids with two α-helices against an antiparallel β sheet), K homology (KH, a highly conserved protein domain of approximately 70 amino acids that interacts with either ssRNA or ssDNA), arginine–glycine–glycine (RGG) motifs, zinc-finger motifs (ZnF; a family of proteins averaging 30 aa with a ββα topology), and DEAD/DEAH box helicase [55,56]. The combination of consecutive RRMs in an RBP increases binding affinity and specificity. In this model, we propose that the N protein is the scaffold protein which seeds a quadriplex structure in which TDP-43 is the client protein.

These observations have broader implications for the putative role of the interaction of the N protein and RBPs in the genesis of pathological biomolecular condensates in neurodegeneration. RBPs play a critical role in mRNA biogenesis, including transcription, pre-mRNA processing, localization, translation, and decay [48]. RBPs share a core set of common features that may contribute to their propensity to participate in pathological inclusion formation, including their enrichment in IDRs, significantly more so than other human proteins. Within any given RBP, using prediction algorithms, the RBD is more likely to be structured (a lower predicted ID of 0.03–0.05) compared to the non-RBD regions (a higher predicted ID of 0.30–0.40) and to be enriched with post-translational modifications [57]. *Cis*-acting mRNA elements are not unique to an individual *trans*-acting RBP, and thus, multiple mRNA partners for a single RBP and multiple RBPs can interact with a single RNA (through multiple *cis*-acting elements).

Our study does not account for the significant impact that post-translational modifications (PTMs) of either the N protein or TDP-43 will have upon the dynamics of biomolecular condensate formation. Using mass spectroscopy on in vivo cross-linked RNA–protein interactors, it has been shown that RBPs are highly modified by PTMs as a class of proteins and significantly more modified compared to those observed for the larger human protein database [54]. Given that RBPs are also significantly more intrinsically disordered that other protein classes and thus more flexible (or less rigid), this suggests that PTMs contribute to the dynamic regulatory roles of RBPs. In addition, a single RBP may contain multiple RBDs which can assist in coordinating and enhancing mRNA binding. Moreover, the presence of an RBD and IDR in an RBP increases the propensity for LLPS. The characteristics are readily observed in RBPs that are commonly seen to be incorporated in pathological biomolecular condensates found in a broad range of neurodegenerative disorders, such as the RBPs TDP-43, FUS/TLS, EWS, and RBM45 that are key to the pathological NCIs of ALS [58,59]. The crucial role of PTMs in the replication and pathogenicity of the N protein, and in particular, mutations of residues 203–205 of the serine-rich domain, in addition to modifying the phosphorylation state, has recently been described [60]. Future studies will need to address these key modifications to further our understanding of how SARS-CoV-2-associated mutations will impact pathological LLPS. It is also unknown whether the N protein undergoes deamination, as has been reported with the SARS-CoV-2 S protein [61,62], a process which might adversely affect its interaction through the N-CTD.

The key driving forces for LLPS are interactions between aromatic and +ve charged amino acids (i.e., lysine, arginine, and histidine), while glycine enhances fluidity, with glutamine and serine promoting hardness [50]. Proteins that undergo LLPS tend to harbor intrinsically disordered regions (IDRs) or coiled-coil domains which increase self-interaction and aggregation at higher concentrations. IDRs are composed of long stretches of low amino acid diversity lacking in hydrophobic residues (low complexity domains; LCDs) which typically mediate cooperative folding [63]. Although occasionally referred to as prion-like domains (PrLDs), PrDLs are most appropriately considered a subset of IDRs with similarities to yeast prion proteins, with enrichment in glutamine and/or asparagine residues. Typically, amino acids found within LCDs include polar (glutamine and serine) and aromatic (tyrosine) residues. Tyrosine residues generally combine with glycine and serine in forming [G/S] Y [G/S] motifs, which have a tendency to form aggregates in vitro, lack well-defined 3D structures, and favor the formation of hydrogels [48,64]. IDRs also generally consist of repeats of arginine/serine (RS repeat), arginine/glycine (RGG box), arginine, or lysine-rich patches (R/K patches).

While IDRs are the main drivers of LLPS, there are examples of non-prion-like domains that drive LLPS. IDRs can also be the sole RBD in an RBP. Because of their lack of structure, IDRs can coordinate RNA binding in concert with other domains and demonstrate a range of specificities from high to nonselective and may promote protein–RNA co-folding upon interaction with target RNAs [52,64]. At low RNA:RBP stoichiometry, the RNAs will facilitate RBP phase separation, whereas high RNA:RBP stoichiometry leads to soluble RNP formation [65]. Of note, proteins that are highly prone to condensate formation possess both RBDs and IDRs and can act as nucleating centers for increased condensate formation and potentially pathological liquid–solid phase separation (LLPS), thus giving rise to the hallmark nuclear and cytoplasmic inclusions of many neurodegenerative disorders. Biomolecular condensates tend to be enriched in proteins that have dual RNA binding ability and IDRs, with an increased number of RBDs correlating with a greater propensity for phase separation [49].

In the context of pathological biomolecular condensate formation, it is noteworthy that a number of RBPs lack a classical RBD but instead interact with RNA through an IDR. There is increasing evidence that IDRs can drive RNA/RBP interactions with a higher affinity to RNA for RBPs that contain ordered RBDs and which can transition to structured domains upon RNA binding [56]. As such, RBPs participate in LLPS through RBDs, IDRs, and (as will be discussed) an array of post-translational modifications (PTMs).

### 3.2. RBP Post-Translational Modifications as Modulators of Biomolecular Condensates

PTMs are important for tuning intermolecular interactions through regulating the charge state of IDPs and thus, while being capable of regulating LLPS or biomolecular condensate genesis, can also promote liquid–solid phase shifting (LSPS) and “molecular hardening” [50]. PTMs can lead to the weakening or enhancement of multivalent interactions in biomolecular condensates and, through either recruiting or excluding macromolecules to the complex, modulate the physical state of the complex. The most well-characterized RBP PTMs are those related to cellular distribution and interactions, particularly with RNA through modulating RNA binding properties, such as phosphorylation, acetylation, arginine methylation, and sumoylation, while ubiquitinylation is involved with degradation and turnover, with phosphorylation and methylation being key for RNA interactions (with potential opposing interactions) [55]. RBP PTMs influence RNA biogenesis by impacting subcellular localization (e.g., nuclear import), stability, degradation, and translation (through regulating alternative splicing or polyadenylation), modulating the interactions with RNA and other proteins, and modulating their propensity for LLPS (an example is that methylation may preclude phosphorylation at an adjacent site by steric hindrance) [55]. An important role of RBP PTMs is increasingly evident across a range of human diseases, including cancer and neurodegeneration, with relevance to ALS for TDP-43, FUS/TLS, and hnRNP-A/B [50].

### 3.3. Evidence That the N Protein Can Be Involved in Pathological Biomolecular Condensate Formation Relevant to Human Neurodegenerative Disease States

The postulate that a viral infection may set in a motion a cascade of events that can give rise to, accelerate, or be associated with an increased prevalence of neurodegenerative processes is not entirely novel [66,67,68].

Neuropathological studies of individuals dying during the acute manifestations of COVID-19 or shortly thereafter demonstrate prominent neuroinflammatory changes with the presence of SARS-CoV-2 RNA or the expression of either S or N proteins being inconsistent and, when present, not clearly correlated with the severity of the neuroinflammatory pathology [69,70,71,72]. Neuropathological features of hypoxic–ischemic injury often accompanied by vascular pathology with infarction alongside evidence of immune-mediated microvascular pathology and blood–brain barrier compromise in the acute stages of COVID-19 have been well documented [73,74,75,76]. The presence of diffuse microglial activation in the brainstem, including microglial nodules, astrogliosis, and perivascular inflammation with parenchymal CD3+ and CD8+ T cells, further supports the concept that the acute neurological manifestations of COVID-19 are the consequence of a prominent neuroinflammatory process in the absence of robust evidence of ongoing SARS-CoV-2 infection of the brain [72,77,78,79].

The question of whether the SARS-CoV-2 virus or N protein can chronically be expressed in the human central nervous system, and more specifically in neurons as opposed to non-neuronal cell populations, remains entirely uncertain. In vitro studies using the OC34 strain of the human coronavirus (HCoV-OC43) in primary murine hippocampal and cortical neuron-enriched cultures demonstrated that neurons were preferentially targeted by the virus, although at later time points, astrocytes were also infected [80]. Infected neurons had largely disappeared by day 7 post-infection. Human IPSC-derived motor neurons have been shown to be susceptible to SARS-CoV-2 infection, and although the levels of viral replication were low, infection could be passaged to VeroE6 cells using a supernatant derived from the infected IPSC-derived motor neurons [81]. Of specific relevance to our work, N protein expression was detected in SARS-CoV-2-infected neurons. In vivo studies using intracerebral inoculums in BALB/c mice were associated with progressive motor dysfunction with prominent neuronal degeneration. Macaques examined at 5 to 6 weeks after SARS-CoV-2 inoculation and thus beyond the acute stages of infection demonstrated an ongoing neuroinflammatory response with microglial nodules, microglial activation, meningeal inflammation, and T-cell infiltrates within the brain parenchyma [82]. Active viral replication at this time point was not observed. Of importance to this discussion, α-synuclein aggregates in the ventral midbrain of six of the eight macaques studied were also observed. Although this latter study did not examine the colocalization of α-synuclein aggregates with either SARS-CoV-2 or the N protein, the ability of both the S1 and N proteins (in particular the N-CTD domain) to interact with α-synuclein, the main component of Parkinson’s disease-associated Lewy bodies, and accelerate pathological fibril formation has been demonstrated by both bioinformatics approaches and in vitro [83,84,85].

It remains to be resolved whether the N protein can be chronically expressed in the human brain or spinal cord and, in doing so, be available to either initiate or accelerate pathological biomolecular condensates with candidate RBPs. Although PCC persistence of SARS-CoV-2 RNA and both the S and N proteins has been documented, including in plasma neuron-derived exosomes for upwards of 16 months post-COVID-19, there are no studies correlating this with features of neurodegeneration [86,87]. There remains controversy as to whether the SARS-CoV-2 virus can directly infect human neurons or whether the neurological complications of virus exposure are the cumulative impact of multiple pathological processes. Recent in vitro evidence supports that the SARS-CoV-2 virus can directly infect astrocytes which then function as the primary viral reservoir capable of mediating neuronal injury [88].

In the context of a potential role of the N protein in inducing pathological oligomers with ALS-associated RBPs, as demonstrated here and elsewhere, there is clear ex vivo evidence that such a process can occur. It is noteworthy that the protein interactome of the N protein is enriched for proteins associated with SGs, including several (e.g., G3BP1/2, Stau1 and Caprin1) that are associated with the pathophysiology of ALS [27,89,90,91,92,93]. Whether pathological LLPS or LSPS driven by the N protein occurs in the human nervous system and can be incriminated in the proposed increase in prevalence of neurodegenerative diseases post-COVID-19 awaits further evidence from both longitudinal case–control studies and detailed neuropathological analysis. Many questions remain before the validity of such a hypothesis can be addressed, including whether there is a natural cellular ‘reservoir’ for SARS-CoV-2 or the N protein within the central nervous system; if such a reservoir exists and is non-neuronal, how the N protein might be propagated into neurons, and having done so, whether it can then propagate pathological biomolecular condensates; further, whether such propagation is directly linked to the development of a chronic degenerative disease state.

## 4. Conclusions

It has been well established that the SARS-CoV-2 N protein shares many features with classical RBPs and in doing so can form both homopolymeric structures in addition to heteropolymeric oligomers as drivers of biomolecular condensates. Using a wide range of methodologies, including in silico projections, ex vivo studies, in vitro cell culture methodologies, and in vivo experimental paradigms, there is substantive evidence that the N protein can undergo pathological LLPS in which endogenous RBPs are incorporated into biomolecular condensates (Figure 5). However, it is less clear whether these observations can be extended into a causation of chronic neurodegeneration in which the hallmark is the presence of pathological neuronal and glial cytoplasmic inclusions and in which the N protein would be postulated to be a major driver of pathological biomolecular condensates. Answering this question in the wake of the COVID-19 pandemic is a critical next step to understanding how to mitigate the projected increase in these disease states, including ALS.

## 5. Materials and Methods

### 5.1. Protein Aggregation Assay

Protein aggregation in the presence of the SARS-CoV-2 N protein, TDP-43 proteins, and short RNA oligos was assessed with the PROTEOSTAT^®^ Protein Aggregation Assay (Enzo (Farmingdale, NY, USA)). Commercially available N protein (GenScript (Piscataway, NJ, USA)) was concentrated ~10-fold and the storage buffer exchanged for 1 × PBS using Pierce™ Protein Concentrators PES 10K (ThermoScientific (Waltham, MA, USA)) according to the manufacturer’s directions. TDP-43 proteins used in this experiment included GST-tagged full-length TDP-43 (consisting of amino acids 1–414 of TDP-43) (generously donated from Emanuele Buratti’s group) or N-terminus TDP-43 (consisting of amino acids 1–269), including both the RRM1 and RRM2 RNA binding domains, purified as previously published [94]. Frozen TDP-43 aliquots were concentrated and the storage buffer exchanged to 1 × PBS similarly to the N protein. Final protein concentrations were determined by BCA microplate assay with BSA as standards. PolyrG, polyrA, and polyrAUG RNA oligos used in the aggregation assays were synthesized by MilliporeSigma (Burlington, MA, USA) and consisted of the sequences 5′-GGGGGGUGGGGGUGGGGG-3′, 5′-AAAAAAAAAAAAAAAAAA-3′, and 5′-GUGUGAAUGAAU-3′, respectively. PROTEOSTAT^®^ detection reagent was mixed according to the manufacturer’s instructions for the volume required. Proteins and/or RNA were gently mixed at the concentrations and molar ratios indicated in turbidity/crowding buffer consisting of Tris-HCl, NaCl, 0.5 mM DTT, and 10% dextran; then, 98 μL of each reaction mixture was quickly mixed with 2 µL of detection reagent pre-loaded in a 96-well black-walled microplate in duplicate. Positive and negative aggregation standards were included and the buffer only was used as a control. Microplates were read with an excitation setting of 550 nm and an emission of 600 nm every 2 min for 50 min in a BioTek (Winuskey, VT, USA) Synergy H1 microplate reader.

### 5.2. Surface Plasmon Resonance

Protein interactions were assessed using a Reichert 2SPR, SR7500DC System. Standard amine coupling (EDC/NHS chemistry) was used to capture commercially available recombinant SARS-CoV-2 N protein (Genscript) on a carboxymethyl dextran hydrogel sensor chip (Reichert (Buffalo, NY, USA)). The concentration of the N protein used for immobilization was 100 µg/mL and the amount of ligand immobilized was approximately 5000 µRIU. TDP-43 analyte proteins (GST-tagged full-length TDP-43, the N-terminus domain containing both the RRM1 and RRM2 domains, and an untagged full-length TDP-43) were dialyzed to running buffer and serially diluted to the concentrations indicated in running buffer (10 mM Hepes pH 7.5, 150 mM NaCl, and 0.1% Tween-20). GST-tagged full-length TDP (FL-TDP-43) was provided as frozen purified aliquots from E. Buratti as described above and the N-terminus TDP-43 was purified by our lab as previously described [94]. The untagged full-length TDP-43 used (Bio-Techne (Minneapolis, MN, USA)) is commercially available and was previously used for SPR interaction experiments [95]. Analyte concentrations of 0.1–1 μM FL-TDP-43, 5–20 μM N-terminus TDP-43 (TDP-43^1–269^), and 0.01–0.5 μM untagged FL-TDP-43 were injected on both the ligand and reference channels at 35 μL/min for 2 min, with a 3 min dissociation time, at 22 °C. The GST protein (2, 10 and 20 µM) and the untagged TDP-43 protein were injected to evaluate possible unspecific binding of the GST tag on full-length TDP-43 to the immobilized N protein (note that the N-terminal TDP-43 protein does not have a GST tag). Buffer injections were used as blanks for the experiments. Sensorgram analysis and binding affinity (dissociation constant; K_D_) calculations were performed using steady-state analysis with the Reichert SPRAutolink (version 1.1.16), TraceDrawer (version 1.8.1), and GraphPad Prism 9.5 software packages.

### 5.3. In Silico Modeling of the N Protein, TDP-43, and RNA Interactions

In silico molecular modeling of the N protein, TDP-43, and RNA interactions was conducted using AlphaFold 3 to generate projections of N protein heteropolymers [43]. AlphaFold input parameters included four copies of the full-length sequence for TDP-43 (Uniprot accession code: Q13148) and two copies of the full-length sequence for the N protein (Uniprot accession code: P0DTC9), where the highest ranking models were selected for further investigation. Ribonucleoprotein complex formation was simulated under these parameters alongside two copies of a 70-mer RNA oligo (5′-GUCGCAGAUUAUGGGUUUACGAUCCGACCUACAAGUUCCGGCCCCCUGUGGUAACCCGCACGCAUUAGGU-3′), derived from Molbiotools random sequence generator (https://molbiotools.com/randomsequencegenerator.php (accessed on 30 May 2024)) with the input RNA parameters specifying a length of 70 and 50% for GC content. PBD files for TDP-43-N protein heteropolymers were then examined via PISA (http://www.ebi.ac.uk/pdbe/prot_int/pistart.html (accessed on 24 July 2024)) to analyze TDP-43 intramolecular interaction properties [96].

## Figures and Tables

**Figure 1 ijms-25-08779-f001:**
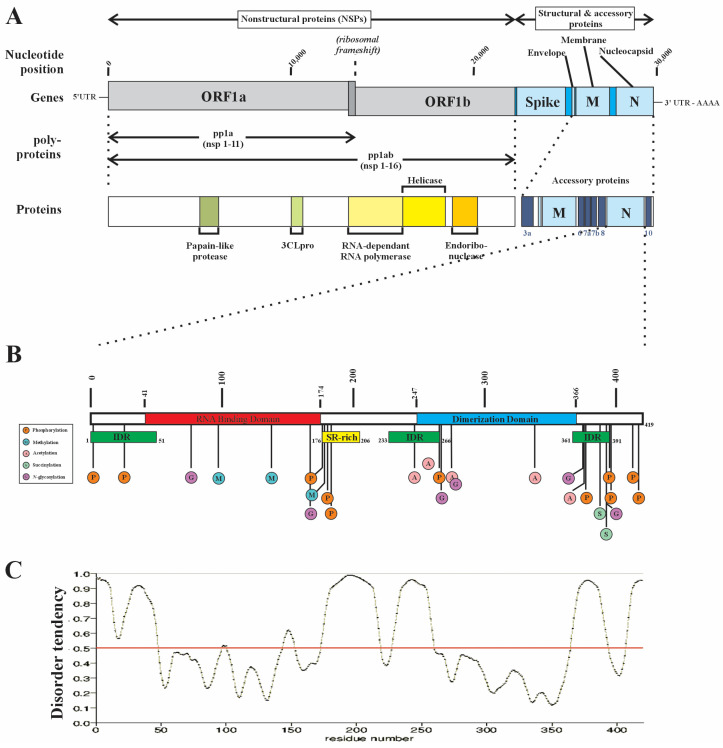
A schematic representation of the SARS-CoV-2 nucleocapsid protein (N protein). (**A**) The structural protein segment of the SARS-CoV-2 gene encodes for multiple proteins, including the spike (S1 and S2 subunits), envelop (E), membrane (M), and nucleocapsid (N) proteins. (**B**) The N protein, encoded by the 9th ORF, is a 419 aa protein of 46 kDa composed of three domains: the N-terminus region contains a prominent RNA binding domain (RBD), a C-terminus dimerization domain that facilitates N protein dimer formation, and a Ser/Arg-rich central flexible linker region (LKR). The NCP undergoes extensive post-translational modification (PTM), predominantly within IDRs. Note the relative paucity of PTMs in the RBD—a feature not commonly observed in the RBPs of ALS-associated RBPs. Methylation at R95 is required for stress granule formation, while F17 has been shown to be the critical interactor for G3PB1 [23,27,28]. (**C**) The N protein contains three intrinsically disordered domains, including the N and C termini and the LKR (intrinsic disorder tendency predicted using PrDOS (protein disorder prediction system (https://prdos.hgc.jp/cgi-bin/result.cgi?ppid=387405p1d1696958772 (accessed on 10 October 2023)) and validated with UniProt (https://www.uniprot.org/uniprotkb/A0A2D1PX79) (accessed on 10 October 2023)). Modified from Strong, 2023 [1].

**Figure 2 ijms-25-08779-f002:**
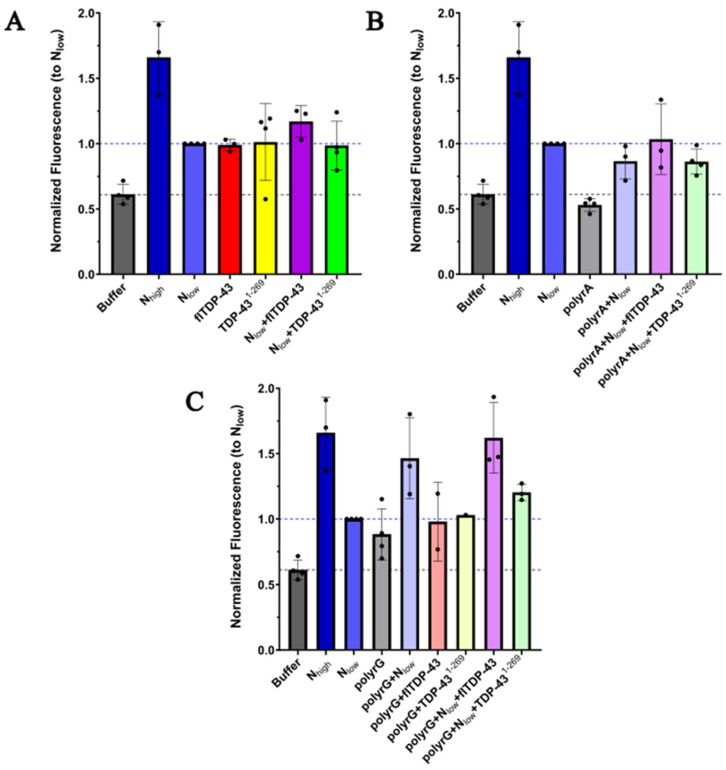
The relative aggregation of purified N and TDP-43 proteins in the absence of RNA (**A**), in the presence of polyrA (**B**), or in the presence of polyrG (**C**). Proteins and/or RNA were mixed at the same time in turbidity buffer with crowding reagent (10% dextran) and added to diluted Proteostat^®^ aggregation detection reagent. The fluorescence shown was recorded with a plate reader approximately 30 min after initial mixing. N was mixed with the aggregate reagent at a high concentration (20–24 µM) or a low concentration (1–2.5 µM). N and full-length TDP-43 (flTDP-43) or N-terminus TDP-43 (TDP-43^1–269^) were mixed at a molar ratio of 1:0.8. N was mixed with the RNA indicated at a molar ratio of 1:0.25. The same molar ratios were used in mixtures containing N, RNA, and TDP-43 variants. The fluorescence was normalized to the low N concentration (N_low_) used in each experiment. The data include on average three independent experiments with black dots showing these values.

**Figure 3 ijms-25-08779-f003:**
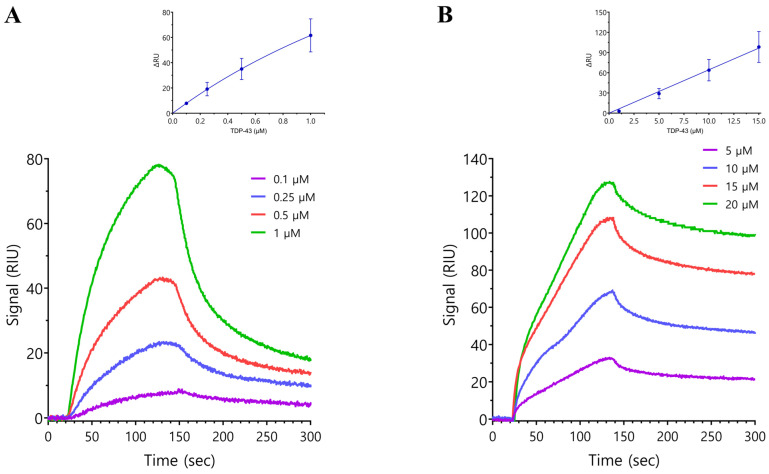
Representative sensorgrams showing the interaction of FL-TDP-43 (**A**) or TDP-43^1–269^ (**B**) with the immobilized SARS-CoV-2 N protein. Protein interactions were assessed by surface plasmon resonance using a Reichert 2SPR, SR7500DC System. The recombinant SARS-CoV-2 N protein was captured on the chip, TDP-43 analyte proteins were serially diluted in running buffer, and sensorgrams were determined at defined concentrations in at least three experiments. The equilibrium constant analysis using non-linear regression is shown with insets above the representative sensorgrams. The binding affinity (K_D_) values using steady-state analysis for the interaction with FL-TDP-43 and TDP-43^1–269^ were 4.88 ± 1.13 μM and >167 μM, respectively, suggesting that the interaction between FL-TDP and the N protein is not mediated through the N-terminus domain of TDP-43.

**Figure 4 ijms-25-08779-f004:**
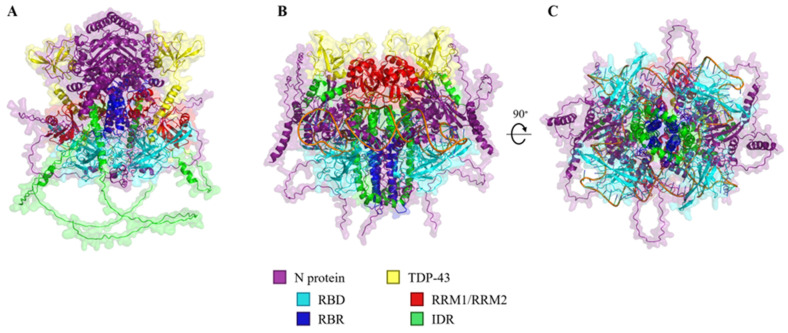
In silico projections of the N protein: TDP-43 heteropolymer complex in the absence or presence of an RNA oligo. (**A**) Four N proteins are modeled in a complex with two FL-TDP-43 proteins. The N protein is displayed in purple, with the N-terminal RNA binding domain (RBD: residues 41–174) in cyan and the C-terminal RNA binding region (RBR: residues 210–246) in blue. TDP-43 is displayed in yellow, with the RRM1/RRM2 (residues 104–176 and 192–262) in red and the C-terminal intrinsically disordered region (IDR: residues 263–414) in green. (**B**) The same complex is modeled in the presence of two random 70-mer RNA oligos. (**C**) With a 90° rotation along the *x*-axis, this displays the bottom view of the heteropolymer. Note the more compact structure in which the IDR of TDP-43 becomes incorporated in proximity to the N-CTD.

**Figure 5 ijms-25-08779-f005:**
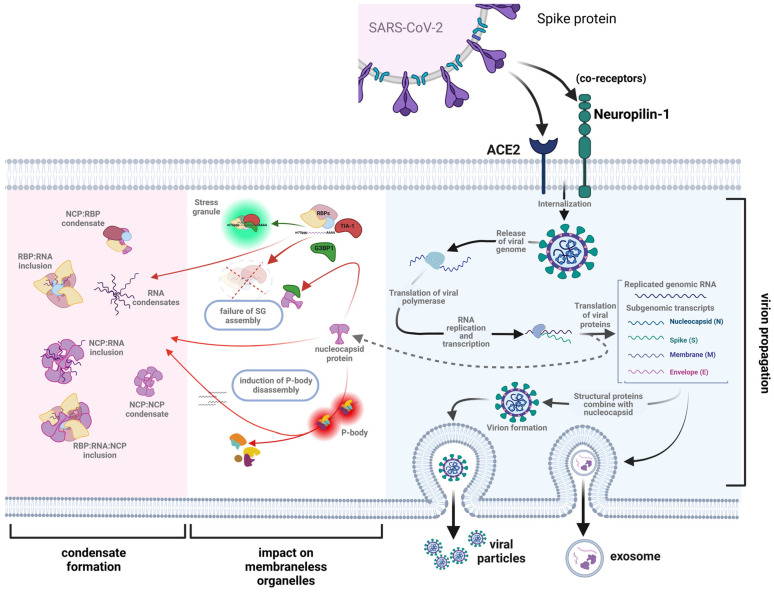
A schematic illustration of the role of the SARS-CoV-2 nucleocapsid protein (N protein) in driving pathological biomolecular condensate formation. Upon endocytosis of the SARS-CoV-2 virus and release of the viral genome, the N protein plays a critical role in the translation of the viral gRNA and the formation of the virion. The latter is released either through exocytosis or encapsulated in exosomes. However, the free N protein also inhibits the formation of stress granules (SGs) by directly interacting with G3BP1, which is critical to the initial assembly of the SG. Not shown is the proposed additive role of the N protein in leading to the disassembly of processing bodies (PBs). The net effects are the formation of pathological biomolecular condensates of varying composition: N protein homopolymers; N protein: RNA heteropolymers; N protein: RBP heteropolymers; and N protein: RNA: RBP heteropolymers. It is hypothesized that any or all of these pathological molecular condensates would play a critical role in the pathogenesis of the neuronal or glial cytoplasmic inclusions that are the neuropathological hallmark of a broad array of neurodegenerative disorders (figure modified from Strong, 2023 [1] and created with BioRender.com).

## Data Availability

The original contributions presented in this study are included in the article/Appendix A; further inquiries can be directed to the corresponding author.

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
