# Peer review of "Phase Separation of SARS-CoV-2 Nucleocapsid Protein with TDP-43 Is Dependent on C-Terminus Domains"

_ijms, 2024, doi:10.3390/ijms25168779_

Round 1

Reviewer 1 Report

Comments and Suggestions for Authors

In this paper, the authors have illustrated that the N protein might form biomolecular condensates with TDP-43 and that this was dependant on the N protein C-terminus do-main (N-CTD) and the intrinsically disordered C-terminus domain of TDP-43. In addition, the manuscript also has the following issues.

1.     In the part of protein aggregate formation test, non-denaturing gel electrophoresis WB and dot blot are required to further verify the formation of aggregates.

2.     What is the basis for selecting and designing polyrA and polyrG sequences? Have you tried other different types of sequences? What was the result?

3.     When verifying the interaction between N protein and TDP-43, CO-IP assay is needed for validation.

4.     The results of the interaction between GST and N protein need to be included in the supplementary materials section.

5.     Further validation of the interaction at the cellular level is required by transfecting plasmids expressing corresponding proteins and different RNA sequences.

Author Response

  1. In this paper, the authors have illustrated that the N protein might form biomolecular condensates with TDP-43 and that this was dependant on the N protein C-terminus domain (N-CTD) and the intrinsically disordered C-terminus domain of TDP-43. In addition, the manuscript also has the following issues.

    1. In the part of protein aggregate formation test, non-denaturing gel electrophoresis WB and dot blot are required to further verify the formation of aggregates.

    Response: We thank the reviewer for this suggestion. Given the limited quantities of protein available, we attempted to use a native gel to further demonstrate the formation of aggregates. Unfortunately, we did not have success with this approach for technical reasons and elected to use the remaining protein within the time window for responding to reviewers to increase our replicate numbers in the accompanying studies.

    1. What is the basis for selecting and designing polyrA and polyrG sequences? Have you tried other different types of sequences? What was the result?

    Response: We have provided references (line 133-134) to the use of polyrA and polyrG in the phase separation assay in which previous authors demonstrated the ability of either to enhance the rate of N protein LLPS. In doing so, we wished to ensure comparability of our results to those that are existent (although not specifically for the interaction of N protein with TDP-43).  With respect to use of other types of sequences, we undertook studies using a polyAUG RNA oligo and have included this in the manuscript (lines 131, 167 – 169, 496; Supplementary figure 1A).  Results with this oligo did not differ from the polyrA oligo.

    1. When verifying the interaction between N protein and TDP-43, CO-IP assay is needed for validation.

    Response: We thank the reviewer for this suggestion which also links directly to comment #5. We agree that these experiments will be a necessary next step, alongside in vivo studies.  However, they are planned for the next series of studies.

    1. The results of the interaction between GST and N protein need to be included in the supplementary materials section.

    Response:  This information has now been added as supplementary figure 2A (cited line 191). This was further supported by demonstrating that an untagged FL-TDP-43 protein showed robust interaction with N protein (also Supplementary figure 2B)

    1. Further validation of the interaction at the cellular level is required by transfecting plasmids expressing corresponding proteins and different RNA sequences.
    Response:  Please see comment #3 response.  We agree and will undertake these in the next series of studies.

Reviewer 2 Report

Comments and Suggestions for Authors

The authors showed that the N protein can form biomolecular condensates with TDP-43 and that this is dependant on the N-CTD and the intrinsically disordered C-terminus domain of TDP-43. The topic is important and interesting. The authors demonstrated that the C terminus domain of TDP-43 is the main determinant of the interaction between N protein and TDP-43, but only full length TDP-43 and N-terminus were shown in the figure 2. Any results using C-terminus?

Suggestion 2, please move line 421 to 432 to discussion section as all these lines are questions to be answered, not the conclusion of this study.

Author Response

The authors showed that the N protein can form biomolecular condensates with TDP-43 and that this is dependant on the N-CTD and the intrinsically disordered C-terminus domain of TDP-43. The topic is important and interesting. The authors demonstrated that the C terminus domain of TDP-43 is the main determinant of the interaction between N protein and TDP-43, but only full length TDP-43 and N-terminus were shown in the figure 2. Any results using C-terminus?

Response:  We thank the reviewer for this comment but unfortunately, we were unable to maintain the C-terminus fragment in solution outside of a urea-based buffer during its dialysis into the appropriate buffers for either SPR or protein aggregation studies. While we had indicated this for the SPR analysis in the previous version (lines 193-195), we have similarly addressed this in the aggregation studies by adding line 169-171 to the text.

Suggestion 2, please move line 421 to 432 to discussion section as all these lines are questions to be answered, not the conclusion of this study.

Response: We thank the reviewer for this suggestion and have modified the final paragraph of the discussion and the conclusion paragraph accordingly.

Reviewer 3 Report

Comments and Suggestions for Authors

Thank you very much for the opportunity to review your manuscript entitled “Phase separation of SARS-CoV-2 nucleocapsid protein with 2 TDP-43 is dependant on C-terminus domains”. In summary, the manuscript describes the use of protein sequence and structure bioinformatics, surface plasmon resonance, and a protein aggregation assay to determine the ability of the SARS-CoV-2 N protein to interact with TDP-43, particularly in a RNA-dependent manner. The work is very nicely and thoroughly described, thank you for the great read. The Discussion section comprehensively addresses all potential limitations of the study. The figures are also very nicely presented and are easy to understand. I just have a couple of very minor questions and suggestions for improvement. Any consideration in these points would be greatly appreciated. I look forward to seeing this published soon. Congratulations on the great work!

Results

1) Is it possible to look at the affinity of the N protein and TDP-43 with and without the presence of RNA using SPR? Otherwise, how might the kinetics be affected?

2) Could an extra row of images in Figure 4 be presented to better visualize the critical protein-protein and protein-RNA interactions. Also to show the differences between the models with and without RNA oligomers? The overview images are helpful but not clear.

3) Based on the Alphafold3 model, are there any N protein mutations identified in the variants of concern (e.g. Delta, Omicron) that may alter the ability of the N protein to aggregate or interact with TDP-43.

Discussion

4) Considering that the “IDRs also generally consist of repeats of arginine/serine (RS repeat), arginine/glycine (RGG box), arginine or lysine-rich patches (R/K patches)”, has the N protein been shown to undergo asparagine deamidation as the spike protein has (PMIDs: 34499924, 37178506)? If not, do the authors consider that this may affect N protein function? Please cite the two references.

Author Response

Thank you very much for the opportunity to review your manuscript entitled “Phase separation of SARS-CoV-2 nucleocapsid protein with 2 TDP-43 is dependant on C-terminus domains”. In summary, the manuscript describes the use of protein sequence and structure bioinformatics, surface plasmon resonance, and a protein aggregation assay to determine the ability of the SARS-CoV-2 N protein to interact with TDP-43, particularly in a RNA-dependent manner. The work is very nicely and thoroughly described, thank you for the great read. The Discussion section comprehensively addresses all potential limitations of the study. The figures are also very nicely presented and are easy to understand. I just have a couple of very minor questions and suggestions for improvement. Any consideration in these points would be greatly appreciated. I look forward to seeing this published soon. Congratulations on the great work!

Results

1. Is it possible to look at the affinity of the N protein and TDP-43 with and without the presence of RNA using SPR? Otherwise, how might the kinetics be affected?

Response:  We thank the reviewer for this terrific suggestion. Indeed, we did attempt this. Unfortunately, we could never get a return to baseline in the dissociation phase in the presence of buffer alone and were thus never comfortable that the results were valid. It was specifically very challenging to determine in what sequence to add the individual interactor – we attempted loading the chip with either the N protein or TDP-43 first, then the RNA, and then the other protein. Getting to steady state after adding the RNA never occurred. Adding the RNA in advance to either protein just resulted in aggregates (presumably) blocking the chip.  In the end, we had to move on.

2.  Could an extra row of images in Figure 4 be presented to better visualize the critical protein-protein and protein-RNA interactions. Also to show the differences between the models with and without RNA oligomers? The overview images are helpful but not clear.

Response:  We appreciate this comment and have provided a supplementary figure (3) which demonstrates the complexity of the interaction between the N proteins, TDP-43 and RNA in assembling the proposed pathological complex.  We realize that this does not fully answer the question of the reviewer in defining the critical protein-protein and protein-RNA interactions (and providing a map of such); however this project that is currently underway but proving to be complex and beyond the scope of this current paper. 

3. Based on the Alphafold3 model, are there any N protein mutations identified in the variants of concern (e.g. Delta, Omicron) that may alter the ability of the N protein to aggregate or interact with TDP-43.

Response:  We very much appreciate this question and note that this is the topic of a series of studies that will follow on to this paper.  Specifically, we are interested in the R203K and G204R mutations of the alpha and omicron variants in N protein’s serine rich domain which have been shown to alter not only the phosphorylation state of the protein, but also viral replication – suggesting a fundamental functional change (Johnson et al., PLOS Pathogens, 2022). While other mutations, given the role of the SR region in “weak” RNA binding and its role structurally, are also of interest, we have elected to start with this. Nonetheless, we have raised this in the discussions section, alongside the response to query 4 (line 319-323). 

Discussion

4.  Considering that the “IDRs also generally consist of repeats of arginine/serine (RS repeat), arginine/glycine (RGG box), arginine or lysine-rich patches (R/K patches)”, has the N protein been shown to undergo asparagine deamidation as the spike protein has (PMIDs: 34499924, 37178506)? If not, do the authors consider that this may affect N protein function? Please cite the two references.

Response:  We are unaware of asparagine deamination having been reported for the N protein (including a revised search in the process of writing this response). However, one might predict that this would be possible at residues 276-279 and impact on the interaction between the N-CTD and IDR of TDP-43.  It will be interesting to pursue this, and the suggestions of comment 3, in future studies. In the meantime, we have raised this issue in the discussion, lines 323-325.

Reviewer 4 Report

Comments and Suggestions for Authors

The manuscript ijms-3105775 reports the N protein condensation with TDP-43. I recommend the publication after minor revisions, as follows below:

1) In Figure 3 the authors provided the sensorgrams, however, where are the nonlinear regression curve fits used to determine the kinetic and constant parameters? Additionally, provide the association rate constant (ka or kon), dissociation rate constant (kd or koff), and the equilibrium or affinity constant.

2) The STD-NMR approach is interesting to better understand the binding protein-ligand. Please, explore this technique in this work. The results will validate the in silico data. Please, see DOI: 10.1016/bs.mie.2018.08.018

3) What is the reproducibility of the SPR assays? Please, do this assay at least in triplicate and provide the average sensorgram and the standard deviation in the parameters.

4) The authors conducted in silico modelling; however, they did not apply molecular dynamics simulations to verify the stability of the model. Please, do it and provide the RMSD, RMSF, and gyration radius plots.

Author Response

The manuscript ijms-3105775 reports the N protein condensation with TDP-43. I recommend the publication after minor revisions, as follows below:

  • In Figure 3 the authors provided the sensorgrams, however, where are the nonlinear regression curve fits used to determine the kinetic and constant parameters? Additionally, provide the association rate constant (ka or kon), dissociation rate constant (kd or koff), and the equilibrium or affinity constant.

We appreciate this helpful comment of the reviewer and recognize that we did not make it clear in the text that we had elected to use the steady-state method of calculating the binding affinity values. This was in preference to the kinetic fitting methodology to which they refer given our specific interest in providing data comparable to the cited literature using a similar methodology for the N protein. We have provided greater clarity in the text regarding this point (lines 187, 519).  We have included the non-linear regression equilibrium curves in Figure 3 as requested. Note also that in follow-up studies based on the reviewer’s comments, we were able to repeat the experiments using unlabelled FL-TDP-43 and reached steady state (Supplementary figure 2B) with a KD (steady-state calculation) of 125 +/_ 41.6 nM, further supporting the preferential interaction of full length versus the N-terminus (RRM containing) domain of TDP-43 (cited in line 190 – 191)

2) The STD-NMR approach is interesting to better understand the binding protein-ligand. Please, explore this technique in this work. The results will validate the in silico data. Please, see DOI: 10.1016/bs.mie.2018.08.018

We thank the reviewer for this interesting suggestion and have reviewed the recommended manuscript. While we are not the position to be able to undertake this specific methodology at this point in time, we do agree that additional studies are needed to validate our in silico data and as such have clarified this further in the text as key to future studies. In doing so, we also address a very similar concern raised by Reviewer #4

  • What is the reproducibility of the SPR assays? Please, do this assay at least in triplicate and provide the average sensorgram and the standard deviation in the parameters.

We have indicated that the SPR assays were performed in at minimum 3 experiments each (line 200) and have been clear to indicate that the figures provided are representative of these. The KD values have been revised based on additional replicates and contain standard deviations (line 203).

4) The authors conducted in silico modelling; however, they did not apply molecular dynamics simulations to verify the stability of the model. Please, do it and provide the RMSD, RMSF, and gyration radius plots.

We thank the reviewer for their insightful comments and recommendation to conduct a molecular dynamics study to further validate the predicted protein-RNA complex model. Although such a study would add to the understanding of the protein-nucleic acid interactions, conducting a comprehensive molecular dynamics study, especially for a large and complex system such as the protein-RNA complex in question, requires significant computational resources and time. This process involves setting up the system, running simulations and performing extensive analyses to ensure the results are as accurate and meaningful as possible.  Furthermore, correlating these findings with additional biochemical analyses would extend the time required even further. Considering these constraints, we chose to focus our efforts on the most critical aspects of the manuscript revisions that could be addressed within the deadline. We plan to pursue molecular dynamics studies and further biochemical analysis in future work, where we can allocate the appropriate time/resources to performing a rigorous analysis. The current model is presented as a preliminary analysis that is meant to be suggestive. We are confident that our current findings, supported by the additional experimental data presented, provide a solid foundation for understanding the protein-RNA complex and will be of interest and use to readers.

Reviewer 5 Report

Comments and Suggestions for Authors

The manuscript entitled “Phase separation of SARS-CoV-2 nucleocapsid protein with TDP-43 is dependant on C-terminus domains“ by Strong et Al, present very interesting data about SARS-CoV-2 nucleocapsid protein (N protein), that is essential for viral replication, forming a ribonucleoprotein (RNP) complex to drive viral RNA translation and suppress stress granules and processing bodies, increasing uncoated RNA availability. Authors showed that the N protein forms condensates with TDP-43 via their C-terminal domains, a process accelerated by RNA. They also used in silico modeling to indicate these condensates include an N protein quadriplex with TDP-43's C-terminus incorporated.

Despite the huge interest of the manuscript, I’m little bit perplexed about in silico part. Authors used AlphaFold to create the molecular complex, without checking interacting residues and/or equilibrate the system. AlphaFold is a very useful tool, but it is a black box. As bioinformatician, I strongly recommend to Authors to improve this analysis, by using some other software 

-       Verifying the complex stability (e.g. RMSD, RMSF, geometrical and thermodynamical stability) via molecular dynamics simulations,

-       Computing molecular energies (e.g. potential, total, interactions),

-       Analyzing residues involved in molecular interactions,

Otherwise, I would like to suggest advising the reader that the in silico analysis is a preliminary analysis, also introducing and discussing the weakness of this part.

Material and methods sections should be improved to guide the reader in research reproducibility. 

Author Response

The manuscript entitled “Phase separation of SARS-CoV-2 nucleocapsid protein with TDP-43 is dependant on C-terminus domains” by Strong et Al, present very interesting data about SARS-CoV-2 nucleocapsid protein (N protein), that is essential for viral replication, forming a ribonucleoprotein (RNP) complex to drive viral RNA translation and suppress stress granules and processing bodies, increasing uncoated RNA availability. Authors showed that the N protein forms condensates with TDP-43 via their C-terminal domains, a process accelerated by RNA. They also used in silico modeling to indicate these condensates include an N protein quadriplex with TDP-43's C-terminus incorporated.

Despite the huge interest of the manuscript, I’m little bit perplexed about in silico part. Authors used AlphaFold to create the molecular complex, without checking interacting residues and/or equilibrate the system. AlphaFold is a very useful tool, but it is a black box. As bioinformatician, I strongly recommend to Authors to improve this analysis, by using some other software 

- Verifying the complex stability (e.g. RMSD, RMSF, geometrical and thermodynamical stability) via molecular dynamics simulations,
- Computing molecular energies (e.g. potential, total, interactions),

- Analyzing residues involved in molecular interactions,

Otherwise, I would like to suggest advising the reader that the in silico analysis is a preliminary analysis, also introducing and discussing the weakness of this part.

Response:  We thank the reviewer for these suggestions and have incorporate an additional analysis of the intramolecular interaction properties of TDP-43 with the N protein in the presence or absence of RNA (line 223; a new supplementary Table 1; methods section now includes the PISA methodology employed - line 539-540). In addition to this, we have modified the opening paragraph of the discussion to acknowledge and emphasize that the in silico results are preliminary (lines 256-260; supported by 3 relevant references for the reader).

Material and methods sections should be improved to guide the reader in research reproducibility.

Response:  We trust that the modifications made to the paper in addressing the comments of the reviewers addresses this issue.

Round 2

Reviewer 2 Report

Comments and Suggestions for Authors

this is an updated version and the authors have addressed the concerns.

Reviewer 5 Report

Comments and Suggestions for Authors

Authors addressed all the issues.